# The Parental Stress Scale: Psychometric Properties in Pediatric Hospital Emergency Setting

**DOI:** 10.3390/ijerph20064771

**Published:** 2023-03-08

**Authors:** Néstor Montoro-Pérez, Silvia Escribano, Miguel Richart-Martínez, María Isabel Mármol-López, Raimunda Montejano-Lozoya

**Affiliations:** 1Department of Nursing, Faculty of Health Sciences, University of Alicante, 03690 San Vicente del Raspeig, Spain; 2Research Group GREIACC, Health Research Institute La Fe, Av. Fernando Abril Martorell, 106, Hospital La Fe, 46016 Valencia, Spain; 3Department of Nursing, Faculty of Health Sciences, Institute for Health and Biomedical Research (ISABIAL), University of Alicante, 03690 San Vicente del Raspeig, Spain; 4Nursing School La Fe, Adscript Center of University of Valencia, 46026 Valencia, Spain

**Keywords:** psychometric evaluation, parental stress scale, pediatrics, emergency department

## Abstract

Parental psychological distress has been identified as a predisposing factor in attendance at and the inappropriate use of hospital pediatric emergency departments (PEDs). The aim of the study was to validate the Parental Stress Scale (PSS), a 12-item Spanish scale, in parents seeking care at PEDs. The study involved 270 participants with a mean age of 37.9 (SD = 6.76) years, of which 77.4% were women. The properties of the PSS were analyzed. The scale showed adequate internal consistency for the different factors (0.80 for the “Stressors” factor and 0.78 for the “Baby’s Rewards” factor) and optimal model fit (chi-square = 107.686; df = 53; CFI = 0.99; TLI = 0.98; RMSEA = 0.028; 90% CI = 0.00–0.05). The 12-item Spanish version of the PSS is a valid and reliable instrument for assessing the stress levels of parents seeking care in PEDs.

## 1. Introduction

Bringing a child into the family and parenting is often rewarding for parents, but it can lead to feelings of stress. In fact, the transition and journey towards parenthood is considered by many parents to be one of the most difficult periods of their lives [1]. Parental stress is, therefore, taken to mean the stress that parents feel as a result of the daunting task of rearing and bringing up their children. In particular, parents have the subjective perception that they are unable to cope with the demands of parenting, leaving them feeling ineffective and powerless in fulfilling their parental role and culminating in stressful experiences [2,3].

A number of factors play a role in the onset of parental stress. On the one hand, these include the characteristics of the family, such as unemployment, financial difficulties, marital stress, divorce, single parenthood, and low perceived social support. On the other hand, are child characteristics, such as behavioral, health, and developmental problems [4,5,6]. It has in fact been confirmed that sustained high levels of parental stress over time can lead to poor emotional management of children. This ultimately affects parents’ capacity to handle the tasks and demands of parenting, leading to cognitive developmental problems, attention difficulties, inappropriate prosocial behavior, and insecure attachment [7,8]. This sets in motion a vicious circle, disrupting the quality of life and relationships of both [9].

Meanwhile, the scientific literature shows that psychological distress and high levels of parental stress are associated with elevated demand for care in hospital pediatric emergency departments (PEDs), which is not justified by the actual seriousness of the pathology [10,11,12]. Consequences of the inappropriate use of emergency departments include increased costs, service overload, poor care, staff burnout, user dissatisfaction, and at times even users walking out without having been examined, diagnosed, or treated by a doctor [13]. In this context, effective parental stress assessment tools are, therefore, needed to reduce the frequent and non-emergency visits to such services resulting from high levels of parental stress.

Several tools are available for measuring parental stress, but one of the most widely used internationally is the Parental Stress Scale (PSS), developed by Berry and Jones [2,14,15]. The study for the creation and initial validation of the scale was carried out on a sample of clinical and non-clinical children. The final instrument, following exploratory factor analysis (EFA), consisted of 16 items and four factors: “Parental Stressors”, “Lack of Control”, “Parental Satisfaction”, and “Parental Rewards”. In the EFA, of the initial 18 items, items 2 [There is little or nothing I wouldn’t do for my child (ren) if it was necessary] and 3 [Caring for my child (ren) sometimes takes more time and energy than I have to give] failed to load on any factor, while item 16 (Having children has meant having too few choices and too little control over my life) shared factor loadings with the “Lack of Control” and “Parental Stressors” factors (0.43 and 0.54, respectively). Furthermore, item 18 [I find my child (ren) enjoyable] also shared factor loadings with the “Parental Rewards” and “Parental Satisfaction” factors (0.50 and 0.47, respectively). The final scale that was obtained following EFA had good psychometric properties: internal consistency (α = 0.83), test-retest reliability over a six-week period (r = 0.81; *p* < 0.01), and convergent validity with the Parenting Stress Index (r = 0.75; *p* < 0.01) and the Perceived Stress Scale (r = 0.50; *p* < 0.01) [2]. Several studies have since validated the scale in different languages (Bahasa Malaysia, Brazilian, Portuguese, Chinese, Danish, Hindi, Greek, and Spanish) and different population samples (normative population, parents of children with behavioral and/or developmental problems, sick children, and children with sleeping disorders), obtaining differing factorial structures (1, 2, 3, and 4 factors) and removing certain items because they failed to load substantially on any factor [14,15,16,17].

Mixão et al. [17] validated the instrument in Portuguese in a sample of 416 parents of children ranging in age from one month to 15 years old, who had sought care in the pediatric emergency department of a general hospital. The authors performed an EFA with the original 18 items yielding four factors. The internal consistency that was obtained for the various dimensions ranged from 0.57 to 0.78. In Spain, the only study available is by Oronoz et al. [18], in which the tool was adapted to Spanish with a sample of 411 first-time parents, producing a final scale after EFA of 12 items and two factors: “Baby’s Rewards” and “Stressors”.

However, to the best of our knowledge, there is no valid instrument in Spanish for measuring parental stress in the hospital pediatric emergency setting, hence the importance of this research which aims to examine the psychometric properties of the 12-item PSS in Spanish that was developed by Oronoz et al. [18] in parents of children seeking care in the PED.

## 2. Materials and Methods

### 2.1. Design & Participants

An instrumental study was developed, that was designed for adapting instruments to new contexts and analyzing their psychometric properties [19]. Participants were selected by non-probability convenience sampling at a referral hospital in Valencia (Spain). Selection took place between September 2021 and January 2022 according to the following selection criteria: (1) be a parent of a child seeking care in the hospital’s pediatric emergency department, and (2) speak Spanish fluently. The sample size was calculated using statistical analyses of optimal conditions, which gave a minimum sample size of 200 participants for a confirmatory factor analysis (CFA) [20,21]. The final sample, once dropouts had been eliminated, comprised of a total of 270 people.

### 2.2. Data Collection Tools

For data collection purposes, three documents were merged into one. This included:-An ad hoc questionnaire: prepared specifically for this study and including sociodemographic variables such as: age (as a continuous variable); respondent (father, mother); nationality (Spanish, other); level of education (primary, secondary, higher); marital status (single, married/in a stable relationship, divorced/widowed); family unit (nuclear family, extended family, homoparental family, separated-parent family, blended family); socioeconomic status based on monthly household income (with nine response categories ranging from less than 600 euros to more than 6000 euros per month); age of the child in months (as a continuous variable); number of children in the family unit (as a continuous variable); whether or not the child was the first child (yes, no); and the perceived level of anxiety (with five response categories ranging from 1 = not anxious at all to 5 = very anxious).-Parental Stress Scale (PSS): original instrument developed by Berry and Jones [2]. We used the 12-item Spanish version that was developed by Oronoz et al. [18]. The items are answered using a five-point Likert-type scale (1 = strongly disagree to 5 = strongly agree). A higher score on the scale indicates higher levels of parental stress. As a result of the translation-retrotranslation process and in consensus with experts in the field of parenting, item 16 was eliminated from the scale on the grounds of ambiguity, leaving an initial scale of 17 items. The final scale after EFA consisted of 12 items and two factors (“Stressors” variance = 10.1% and “Baby’s Rewards” variance = 23.4%). The instrument shows good internal consistency for both factors (α = 0.76 for “Stressors” and α = 0.77 for “Baby’s Rewards”). Convergent validity was determined using the State-Trait Anxiety Inventory (STAI) scale and the Beck Depression Inventory (BDI), obtaining statistically significant correlations in the hypothesized direction and magnitude (r = 0.49, *p* < 0.001 and r = 0.51, *p* < 0.001, respectively) [18].-State-Trait Anxiety Inventory (STAI E-7): the 7-item Spanish version that was developed by Perpiñá-Galvañ et al. [22]. This instrument consists of the items that were proposed by Chlan et al. plus item 1 of the original scale that was developed by Spielberger. The questionnaire measures state anxiety. The items are answered using a four-point Likert scale (0 = not at all to 3 = very much). A higher score on the scale indicates higher levels of anxiety. The scale has good internal consistency (α = 0.89) [22]. The ordinal alpha coefficient in our study was 0.91.

### 2.3. Procedure

The parents’ initial contact with PED healthcare staff took place in the triage area. Here the study objective was explained and they were invited to participate. All those wishing to participate were provided with one document containing all the study variables with the PSS and STAI-E7 scales, and a separate document for signing the informed consent form. Once the paperwork was completed, it was placed in a sealed envelope and passed to the lead researcher. Preliminary tests were carried out to check whether the document was difficult for the interviewees to interpret, as well as to ascertain if there were any comprehension issues. After the first 10 participants, it was noted that the document did not present reading comprehension difficulties.

### 2.4. Data Analysis

SPSS Statistics version 21.1 for OX [23] was used for descriptive, normative, and product-moment correlation analyses. To analyze the scale’s psychometric properties, the free software R (version 3.6.4) was used [24]. The assessment instrument’s performance was examined by calculating the skewness and kurtosis of the data and the floor and ceiling effects. The distribution of the variables is considered normal when the skewness and kurtosis coefficients lie between −1.5 and 1.5 [20,21]. According to the literature, floor or ceiling effects occur when more than 15% of the participants’ responses fall in the lower or upper response category ranges, suggesting a reduced ability to differentiate between scores [25]. The data were considered ordinal as per the criteria of Rhemtulla et al. [26]. A CFA was performed. Estimates were obtained using the robust weighted least squares mean and variance adjusted (WLSMV) method, used with ordinal variables [25] from the Lavaan [27] package in R. The model fit to the data was analyzed using the comparative fit index (CFI), Tucker–Lewis Index (TLI), and root mean square error of approximation (RMSEA) [27], with index values > 0.90 and RMSEA < 0.06 [28,29] being considered adequate. Internal consistency was calculated by means of the ordinal alpha coefficient, which is more accurate for categorical response scales. The coefficient α ≥ 0.70 was accepted as an indicator of good reliability [30,31]. Convergent validity was assessed by product-moment correlation between the PSS, the STAI E-7 and the criterion variable of perceived anxiety. Significant positive correlations between 0.2 and 0.5 were expected [32], which would confirm the hypothesis that higher levels of parental stress lead to higher levels of anxiety.

### 2.5. Ethical Considerations

All participants in the study agreed to take part voluntarily. The study was approved by the center’s research ethics committee (Registration No. 2020-486-1). With respect to data confidentiality, privacy and confidentiality were ensured in accordance with Regulation (EU) 2016/679 of the European Parliament and of the Council of 27 April 2016 [33].

## 3. Results

### 3.1. Sample Sociodemographic Characteristics

The sample characteristics can be seen in Table 1. The study involved 270 participants: the mean age of all the participants was 37.9 (SD = 6.76) years; 77.4% were women; 85.2% were of Spanish nationality; and 27.8% were single, 63% were married or in a stable relationship, and 9.2% were separated, divorced, or widowed. The average household socioeconomic level was between 1501 and 3000 euros per month (35.60%). The majority (56.70%) had a higher level of education. Significant sex-based differences were found for most of the sociodemographic variables.

### 3.2. Characteristics Performance and Psychometric Properties of the Scale

Table 2 shows the performance of the scale’s component items. Ceiling and floor effects, skewness, and kurtosis were observed for all the items, so the data were considered ordinal. The analysis was based on a congeneric model taking into account the scale structure that was obtained in the EFA by Oronoz et al. [18]. In our study, the CFA of the questionnaire showed an adequate fit of the data to the structure of the 12-item Spanish version (chi-square = 107.686; df = 53; CFI = 0.99; TLI = 0.98; RMSEA = 0.028, 90% CI = 0.00–0.05). The estimated factor loadings ranged from 0.50 to 0.85 (Figure 1). The ordinal alpha coefficient by dimension was 0.80 for the “Stressors” factor and 0.78 for the “Baby’s Rewards” factor. In terms of convergent validity, scores on the PSS scale’s “Baby’s Rewards” factor correlate positively with the STAI E-7 (r = 0.152, *p* < 0.05). However, no statistical association was found with the “Stressors” factor (r = 0.117; *p* = 0.055). In the criterion variable of perceived anxiety, the PSS correlates positively for both the “Baby’s Rewards” factor (r = 0.218; *p* < 0.01) and the “Stressors” factor (r = 0.197; *p* < 0.01).

### 3.3. Descriptive Analysis of Parental Stress in the Paediatric Emergency Setting

Table 3 shows the mean values of the instrument’s factors. A mean score of 6.52 (SD = 2.35) out of a maximum of 16 was obtained for the “Baby’s Rewards” factor and 17.14 (SD = 6.11) out of a maximum of 35 for the “Stressors” factor. In the analysis by sex, mothers showed higher levels of stress than fathers.

## 4. Discussion

To our knowledge, this is the first study to confirm the 12-item structure of the PSS that was proposed by Oronoz et al. [18] in the PED setting. The CFA shows that the Spanish scale retains the structure determined in the EFA by the authors who validated it in 2007, and the fit indices that were obtained in this study are adequate. Although the original scale that was developed by Berry and Jones [2] consists of four factors (“Parental Stressors”, “Lack of Control”, “Parental Satisfaction”, and “Parental Rewards”), internationally there is inconsistency between the factors that were obtained in the process of adapting the instrument, with two main factors (“Parental Satisfaction” and “Parental Stressors”) being commonly identified [34,35,36]. Only three studies were found in which the original tetra-factorial structure was confirmed with good statistical fit [34,37,38]. This factorial inconsistency could be explained by Pontoppidan et al. [39] and Harding et al. [35], who argue that the tetra-factorial model appears to be a subdivision of two primary factors, whereby the “Parental Stressors” and “Lack of Control” factors make up the “Parental Stressors” factor, while the “Parental Satisfaction” and “Parental Rewards” factors make up the “Parental Satisfaction” factor. An analysis of the scientific literature suggests that the different settings and populations in which the scale has been validated affect the stability of the items and the presence of the different factors [15]. There is, however, no doubt that the available versions accurately measure the construct of parental stress, with those where the items are grouped around two main factors being easier to interpret. In particular, the version that was proposed by Oronoz et al. [18] may be regarded as less invasive and easier to use within the context of this research.

In terms of the performance of the scale and item functioning, a ceiling and floor effect is found for all the instrument’s items, most likely indicating a lack of variability between the different scores, which could make it difficult to adequately discriminate between parents with varying levels of stress [25]. This phenomenon has also been observed in the study by Leung et al. [40] which suggested that the PSS may pose difficulties for parents with low levels of stress. In terms of internal consistency, the indices were optimal and in line with those that were described in the literature [30,31,32], with slightly higher values being identified than those that were obtained in other validation studies of the instrument [18,34,36,41,42,43].

From the perspective of convergent validity, the results show an association between higher levels of stress and anxiety. Our findings are corroborated by several authors who have found that parents with higher levels of stress also have higher levels of anxiety [44,45,46]. In terms of the instruments that are used for convergent validity, the STAI has been widely relied upon in the scientific literature to confirm this psychometric property [18,38,41]. Nevertheless, the values in our study are slightly lower, with one factor identified as having a low correlation and another with no statistical association. This may be explained by the fact that the STAI-E7 only measures momentary anxiety (state), whereas the PSS assesses parental stress over time (trait) [2,18,22]. Although the “Parental Stressors” factor does not correlate with the STAI-E7, the values do come close to being significant, which is confirmed by the criterion variable of perceived anxiety that is included in the study.

With regard to the scores that were obtained for parental stress levels in the PED, our results cannot be compared, as there are no known scientific studies measuring stress levels using the version of the PSS that was proposed by Oronoz et al. [18] in parents seeking care in the PED. This notwithstanding, the data from this study are in line with the literature, with the common phenomenon of mothers having higher levels of stress than fathers across the different study settings [2,15,18].

Finally, the intrinsic limitations of this research must be taken into account when interpreting the results, in particular the ceiling and floor effect that was observed in most of the items, which may make discriminating between the results difficult. However, we have provided percentiles of the instrument in the context studied, thus providing future researchers with useful references to help with the correct interpretation of the results and establish the effectiveness of any future interventions. As far as further lines of research are concerned, there is a need for quantitative studies measuring parental stress in PEDs, given that it has been qualitatively affirmed that parental psychological distress, stress, and anxiety are predisposing factors in the demand for care falling outside the scope of such services [11,12,47].

## 5. Conclusions

The 12-item PSS in Spanish is a valid, reliable, brief, and minimally invasive assessment instrument that is capable of detecting parents with high levels of stress in PED. With knowledge of the stress levels that are experienced by parents, healthcare professionals will be able to develop strategies and interventions to eliminate parental stress, with the ultimate implication of reducing non-emergency visits to the PED caused by these high levels of parental stress.

## Figures and Tables

**Figure 1 ijerph-20-04771-f001:**
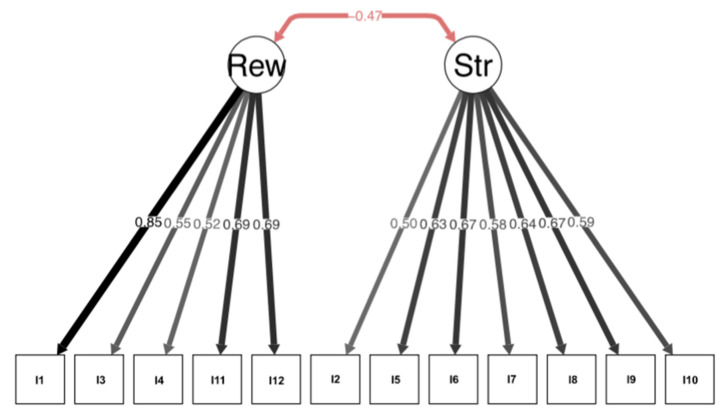
Confirmatory factor analysis of the original structure that was proposed by Oronoz et al. Rew: Rewards; Str: Stressors.

**Table 1 ijerph-20-04771-t001:** Sociodemographic variables of the sample.

Sociodemographic Variables		Totals (*n* = 270) *n* (%)	Statistic *p*-Value
**Age of parents**		37.9 ± 6.76 * 38 (20–54) ^+^	“t” = 5.704 *p* = 0.00
**Parental role *n* (%)**	Father Mother	61 (22.6) 209 (77.4)	*p* = 3.609 ^a^ *p* = 0.013
**Nationality *n* (%)**	Spanish Other	230 (85.2) 40 (14.8)	X^2^ = 15.81 *p* = 0.003
**Level of education *n* (%)**	Primary Secondary Higher	53 (19.6) 91 (33.7) 126 (46.7)	X^2^ = 21.042 *p* = 0.012
**Parent’s marital status *n* (%)**	Single Married/in a stable relationship Divorced/widowed	75 (27.8) 170 (63) 25 (9.2)	X^2^ = 15.81 *p* = 0.003
**Family unit *n* (%)**	Nuclear family Extended family Homoparental family Separated-parent family Blended family	218 (80.7) 20 (7.4) 2 (0.7) 22 (8.1) 8 (3)	X^2^ = 5.376 *p* = 0.251
**Monthly household income *n* (%)**	Less than 600 Euros per month Between 601 and 900 Euros per month Between 901 and 1200 Euros per month Between 1201 and 1500 Euros per month Between 1501 and 3000 Euros per month Between 3001 and 3600 Euros per month Between 3601 and 4200 Euros per month Between 4201 Euros and 6000 Euros per month More than 6000 Euros per month	12 (4.4) 23 (8.5) 33 (12.2) 43 (15.9) 96 (35.6) 35 (13) 20 (7.4) 1 (0.4) 7 (2.6)	X^2^ = 23.324 *p* = 0.003
**Age of child admitted (months)**		58 ± 47.5 * 38 (1–17) ^+^	“t” = 1.862 *p* = 0.832
**N° of children**		1.88 ± 0.996 * 2 (1–7) ^+^	“t” = 0.212 *p* = 0.832
**First child? *n* (%)**	Yes No	151 (55.9) 119 (44.1)	*p* = 0.067 ^a^ *p* = 0.884
**Perceived level of anxiety *n* (%)**	1 (Not anxious at all) 2 3 4 5 (Very anxious)	70 (25.9) 62 (23) 65 (24.1) 39 (14.4) 34 (12.6)	X^2^ = 29.307 *p* = 0.000

(*) Mean ± SD; (^+^) Median (minimum, maximum); X^2^: Pearson’s Chi-square; *p* (^a^) Fisher’s exact statistic; “t”: Student’s *t*-test.

**Table 2 ijerph-20-04771-t002:** PSS Scale and related normative data.

Item	Min	Max	M (SD)	Skewness	Kurtosis	F.E. *n* (%)	C.E. *n* (%)
**1. I am happy in my role as a parent.**	1	4	1.34 (0.72)	2.15	3.79	211 (78.1)	7 (2.6)
**2. Caring for my child(ren) sometimes takes more time and energy than I have to give.**	1	5	3.39 (1.37)	−0.46	−1.03	39 (14.4)	72 (26.7)
**3. I feel close to my child(ren).**	1	4	1.29 (0.62)	2.23	4.61	212 (78.5)	3 (1.1)
**4. I enjoy spending time with my child(ren).**	1	4	1.2 (0.54)	3.19	11.01	230 (85.2)	4 (1.5)
**5. The major source of stress in my life is my child(** **r** **en).**	1	5	2.38 (1.37)	0.65	−0.8	97 (35.9)	33 (12.2)
**6. Having children leaves little time and flexibility in my life.**	1	5	2.65 (1.31)	0.25	−1.03	71 (26.3)	29 (10.7)
**7. Having children has been a financial burden.**	1	5	2.59 (1.34)	0.351	−1.06	77 (28.5)	31 (11.5)
**8. It is difficult to balance different responsibilities because of my child(ren).**	1	5	2.27 (1.25)	0.66	−0.66	96 (35.6)	17 (6.3)
**9. The behaviour of my child(ren) is often embarrassing or stressful to me.**	1	5	2.23 (1.29)	0.77	−0.55	102 (37.8)	19 (7.0)
**10. I feel overwhelmed by the responsibility of being a parent.**	1	5	1.61 (1.01)	1.79	2.57	175 (64.8)	8 (3)
**11. I am satisfied as a parent.**	1	5	1.52 (0.81)	1.55	1.93	173 (64.1)	1 (0.4)
**12. I find my child(ren) enjoyable.**	1	3	1.17 (0.42)	3.47	5.65	229 (84.8)	5 (1.9)

F.E.: Floor effect; C.E.: Ceiling effect; M: Mean; SD: Standard deviation; Min: Minimum; Max: Maximum; Positive items reversed (5 = 1, 4 = 2, 3 = 3, 2 = 4, 1 = 5); Items translated into English for educational purposes only.

**Table 3 ijerph-20-04771-t003:** PSS Scale descriptive statistics.

Factors		Mean (SD)	Range	P25	P50	P75
**Total Sample**	Baby’s Rewards	6.52 (2.35)	5–16	5	5	7
	Stressors	17.14 (6.11)	7–35	13	17	21
**Mothers**	Baby’s Rewards	6.60 (2.43)	5–16	5	5	7
	Stressors	17.51 (6.38)	7–35	13	17	22
**Fathers**	Baby’s Rewards	6.27 (2.05)	5–13	5	5	7
	Stressors	15.86 (4.92)	7–27	12	16	19

SD: Standard deviation; Scores considered with positive items reversed; P25: 25th Percentile; P50: 50th Percentile; P75: 75th Percentile.

## Data Availability

The database can be requested from the lead or corresponding author by formal request from the pertinent institution and MDPI.

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
