# Peer review of "The Parental Stress Scale: Psychometric Properties in Pediatric Hospital Emergency Setting"

_ijerph, 2023, doi:10.3390/ijerph20064771_

Round 1
Reviewer 1 Report
In my point of view, the article can get published after minor revisions. First, to increase the number of the participants. If it possible to investigate parental stress among mothers and fathers separately. Furthermore, the study was conducted during the pandemic era. The authors should comment if parental stress remained unchanged during the pandemic period. In addition, the potential pre-existing mental health of the parent should be mentioned, which could influence both post-traumatic stress symptoms and parenting in ER department.
Additional comments
This article estimates the parental stress scale regarding the psychometric properties in a pediatric ER department. In my point of view, the main question was addressed by the current research as the topic seems to be very interesting in its results, present novelty and originality of statistic research, importance, and impact. The introduction is clear and concise and gives a good rationale and all the figures and tables are easily readable, correct, and informative. Testing parental stress with reliable and valid measurement instruments by specific psychometric tools we can consider evaluating the level of effectiveness of our interventions in family, clinical and/ or preventive settings. Although, in literature are multiple studies showing that the use of fewer or more categories of stress scale may produce clear discrepancies in obtaining sensitive information about the measured psychological trait, so few are published about hospital emergencies settings.
Author Response
REPLY TO REVIEWER 1
The authors would like to thank reviewer 1 for his suggestions. Below is a sectional response to his comments, which we hope will serve as clarification.
"First, to increase the number of the participants"
The sample size was calculated according to the latest statistical recommendations in the field of psychometrics. We provide the reviewer with an excerpt from the latest guidelines: "If someone wants to evaluate the quality of a test, a sample size of at least 200 cases is recommended, even under optimal conditions of high communalities and well-determined factors" (Ferrando et al., 2021; Ferrando-Piera et al., 2022; Lloret-Segura et al., 2014).
"If it possible to investigate parental stress among mothers and fathers separately. Furthermore, the study was conducted during the pandemic era. The authors should comment if parental stress remained unchanged during the pandemic period. In addition, the potential pre-existing mental health of the parent should be mentioned, which could influence both post-traumatic stress symptoms and parenting in ER department"
We understand the reviewer's concern, however, the research we intend to publish is a validation study of a questionnaire and the examination of its psychometric properties. The authors did not intend to assess or evaluate stress levels, the objective was focused simply on confirming the structure of the Parental Stress Scale proposed by Oronoz et al. (2007), never confirmed in the Spanish context, and to see how it performed in the paediatric hospital emergency setting following the guidelines of Carretero-Dios and Pérez (2007). For this purpose, the authors performed a Confirmatory Factor Analysis (CFA), calculated the reliability of the instrument, the hypothesis testing for convergent validity and provided percentiles in the context according to sex and for the total scale, so that future researchers can have a reference and more easily interpret the scores obtained. Although it would be interesting to provide evidence of sensitivity to change of the instrument (understanding that the reviewer refers to it with his statement: "The authors should comment if parental stress remained unchanged during the pandemic period") we do not have repeated samples over time to evaluate and show the results. We consider his contribution important and will take it into account for future lines of research.
“Furthermore, the study was conducted during the pandemic era. The authors should comment if parental stress remained unchanged during the pandemic period”
Regarding the fact that the study was conducted in a pandemic era, it should be noted that in the Spanish context the state of confinement and alarm ended on 21 June 2021, entering the so-called "new normality" (Herrena-Montero, 2022). Furthermore, as we do not intend to assess stress levels, it should be clarified that it is not described in psychometric manuals that the "pandemic or COVID era" affects the content and construct validity of the assessment tools (Medrano & Pérez, 2019).
“This article estimates the parental stress scale regarding the psychometric properties in a pediatric ER department. In my point of view, the main question was addressed by the current research as the topic seems to be very interesting in its results, present novelty and originality of statistic research, importance, and impact. The introduction is clear and concise and gives a good rationale and all the figures and tables are easily readable, correct, and informative. Testing parental stress with reliable and valid measurement instruments by specific psychometric tools we can consider evaluating the level of effectiveness of our interventions in family, clinical and/ or preventive settings. Although, in literature are multiple studies showing that the use of fewer or more categories of stress scale may produce clear discrepancies in obtaining sensitive information about the measured psychological trait, so few are published about hospital emergencies settings.”
The authors would like to thank the reviewer for his gratifying comment.
BIBLIOGRAPHY
Carretero-Dios, H., & Pérez, C. (2007). Standards for the development and review of instrumental studies: Considerations about test selection in psychological research. International journal of clinical and health psychology, 7(3), 863-882.
Ferrando, P. J. (2021). Seven Decades of Factor Analysis: From Yela to the Present Day. Psicothema, 33(3), 378-385.
Ferrando-Piera, P. J., Lorenzo-Seva, U., Hernández-Dorado, A., & Muñiz-Fernández, J. (2022). Decálogo para el Análisis Factorial de los Ítems de un Test. Psicothema.
Herrera-Montero, C. E. (2022). Limitación o suspensión de los derechos en el estado de alarma por la crisis sanitaria del Covid-19: especial referencia a la STC 148/2021, de 14 de julio.
Lloret-Segura, S., Ferreres-Traver, A., Hernández-Baeza, A., & Tomás-Marco, I. (2014). El análisis factorial exploratorio de los ítems: una guía práctica, revisada y actualizada. Anales de psicología/annals of psychology, 30(3), 1151-1169.
Medrano, L. A., Pérez, E., & Fernández, A. (2019). Construcción y adaptación de test psicométricos. Manual de psicometría y evaluación psicológica, 89-100.
Reviewer 2 Report
Thank you for the opportunity to review this study entitled “The Parental Stress Scale: Psychometric Properties in Paediatric Hospital Emergency Setting” (ijerph-2198605).
The research aimed to validate and investigate the psychometric properties of the Spanish PSS (12 items) in parents of children seeking care in the PED. A sample of 270 people was involved in the research.
In my opinion, the research topic is relevant, and the study is interesting. Parallelly, some issues need to be addressed before the paper is suitable for publication.
· Abstract: please avoid reporting indices and references in the abstract.
· Introduction: the psychometric properties of the version by Oronoz et al. (2007) should be explored more in-depth.
· Introduction: in line with the previous comment, hypotheses about the psychometric properties of the PSS (12 items) in parents of children seeking care in the PED should be clearly stated.
· Data collection tools: the Cronbach's alpha of the STAI E-7 in the present sample should be reported in the corresponding section.
· In-text references should be rewritten in line with the IJERPH guidelines: e.g., “Rhemtulla et al. (2012) [26]” should be “Rhemtulla et al. [26].”.
· The “Conclusions” sections should be enriched by highlighting the practical implication of this research (and the utility of using this scale in Spanish Paediatric Hospital Emergency Setting).
Best wishes
Author Response
REPLY TO REVIEWER 2
The authors would like to thank reviewer 2 for his suggestions. Below we respond in sections to his comments and hope that this will serve as clarification.
“Abstract: please avoid reporting indices and references in the abstract”
The authors have taken the reviewer's contribution into consideration and removed the citation from the abstract.
“Introduction: the psychometric properties of the version by Oronoz et al. (2007) should be explored more in-depth”
We understand the reviewer's concern. However, the psychometric properties of Oronoz et al. (2007) are correctly explained in the methods section, which is the most appropriate place according to the latest recommendations (Rosenberg et al., 2013).
We refer to the text of the manuscript: “We used the 12-item Spanish version developed by Oronoz et al. [18]. Items are answered using a five-point Likert-type scale (1 = strongly disagree to 5 = strongly agree). A higher score on the scale indicates higher levels of parental stress. As a result of the translation-retrotranslation process and in consensus with experts in the field of parenting, item 16 was eliminated from the scale on the grounds of ambiguity, leaving an initial scale of 17 items. The final scale after EFA consisted of 12 items and two factors (“Stressors” variance = 10.1% and “Baby’s Rewards” variance = 23.4%). The instrument shows good internal consistency for both factors (α = 0.76 for “Stressors” and α = 0.77 for “Baby’s Rewards”). Convergent validity was determined using the State-Trait Anxiety Inventory (STAI) scale and the Beck Depression Inventory (BDI), obtaining statistically significant correlations in the hypothesised direction and magnitude (r = 0.49, p < 0.001 and r = 0.51, p < 0.001, respectively) [18].”
Reintroducing the information in the introduction section would be redundant, given that the authors aim to give in the introduction section a broad overview of the PSS in the international context.
“Introduction: in line with the previous comment, hypotheses about the psychometric properties of the PSS (12 items) in parents of children seeking care in the PED should be clearly stated.”
The authors tried to respond to the reviewer 2 by clarifying the objective of the research, rephrasing it in a simpler and more concise way: “hence the importance of this research which aims to examine the psychometric properties of the 12-item PSS in Spanish developed by Oronoz et al. [18] in parents of children seeking care in the PED”.
“Data collection tools: the Cronbach's alpha of the STAI E-7 in the present sample should be reported in the corresponding section.”
The authors respond to the reviewer by including the STAI-E7 ordinal alpha of our study, being more accurate than Cronbach's alpha in categorical response scales (Zumbo et al., 2007) and indicated in the latest statistical recommendations (Ferrando-Piera et al., 2022). We attach an excerpt of the text following the reviewer's recommendations: “The ordinal alpha coefficient in our study was 0.91”.
“In-text references should be rewritten in line with the IJERPH guidelines: e.g., “Rhemtulla et al. (2012) [26]” should be “Rhemtulla et al. [26].”
The authors modified the citations in the body of the manuscript following the reviewer's suggestion.
“The Conclusions sections should be enriched by highlighting the practical implication of this research (and the utility of using this scale in Spanish Paediatric Hospital Emergency Setting).”
The authors modified the conclusion based on the reviewer's suggestions by including the practical implications. The following is an extract from the manuscript: “The 12-item PSS in Spanish is a valid, reliable, brief and minimally invasive assessment instrument capable of detecting parents with high levels of stress in PED. With knowledge of the stress levels experienced by parents, healthcare professionals will be able to develop strategies and interventions to eliminate parental stress, with the ultimate implication of reducing non-emergency visits to the PED caused by these high levels of parental stress.”
BIBLIOGRAPHY
Ferrando-Piera, P. J., Lorenzo-Seva, U., Hernández-Dorado, A., & Muñiz-Fernández, J. (2022). Decálogo para el Análisis Factorial de los Ítems de un Test. Psicothema.
Rosenberg, J., Bauchner, H., Backus, J., De Leeuw, P., Drazen, J., Frizelle, F., & International Committee of Medical Journal Editors. (2013). The new ICMJE recommendations. Dan Med J, 60(10), 1-2.
Zumbo, B. D., Gadermann, A. M., & Zeisser, C. (2007). Ordinal versions of coefficients alpha and theta for Likert rating scales. Journal of modern applied statistical methods, 6(1), 4.
Reviewer 3 Report
The theme of the work is relevant. The abstract indicates the objective of the work
The method section is well structured. Especially design, sampling and result parts are consistent with each other.
I think the manuscript meets expectations and is interesting, well elaborated and structured. Some modifications are recommended to increase the final quality:
The quantitative part of the results section needs to be improved. . Confirmatory Factor Analysis of the Original Structure Proposed by Oronoz et al. (2007). was reported but I did not find the CFA of this study. CFA of this study also should be reported.
Author Response
REPLY TO REVIEWER 3
The authors would like to thank reviewer 3 for his suggestions. Below is a response to his comments and we hope that this will serve as clarification.
“The theme of the work is relevant. The abstract indicates the objective of the work. The method section is well structured. Especially design, sampling and result parts are consistent with each other. I think the manuscript meets expectations and is interesting, well elaborated and structured. Some modifications are recommended to increase the final quality: The quantitative part of the results section needs to be improved. Confirmatory Factor Analysis of the Original Structure Proposed by Oronoz et al. (2007). was reported but I did not find the CFA of this study. CFA of this study also should be reported.”
The authors understand the reviewer's concern, however, the data presented in the results section is the CFA conducted in our study. As indicated in the methods section, Oronoz et al. (2007) perform a Exploratory Factor Analysis (EFA), not a CFA. We attach an extract from the manuscript: “We used the 12-item Spanish version developed by Oronoz et al. [18]. Items are answered using a five-point Likert-type scale (1 = strongly disagree to 5 = strongly agree). A higher score on the scale indicates higher levels of parental stress. As a result of the translation-retrotranslation process and in consensus with experts in the field of parenting, item 16 was eliminated from the scale on the grounds of ambiguity, leaving an initial scale of 17 items. The final scale after EFA consisted of 12 items and two factors (“Stressors” variance = 10.1% and “Baby’s Rewards” variance = 23.4%). The instrument shows good internal consistency for both factors (α = 0.76 for “Stressors” and α = 0.77 for “Baby’s Rewards”). Convergent validity was determined using the State-Trait Anxiety Inventory (STAI) scale and the Beck Depression Inventory (BDI), obtaining statistically significant correlations in the hypothesised direction and magnitude (r = 0.49, p < 0.001 and r = 0.51, p < 0.001, respectively) [18].”
However, the authors tried to clarify this in the manuscript by adding "in our study" in the results section. We attach an extract of our results presented in the results section making the clarification proposed by the reviewer: “Table 2 shows the performance of the scale’s component items. Ceiling and floor effects, skewness and kurtosis were observed for all items, so the data were considered ordinal. The analysis was based on a congeneric model taking into account the scale structure obtained in the EFA by Oronoz et al. [18]. In our study the CFA of the questionnaire showed an adequate fit of the data to the structure of the 12-item Spanish version (chi-square = 107.686; df = 53; CFI = 0.99; TLI = 0.98; RMSEA = 0.028, 90% CI = 0.00–0.05). The estimated factor loadings ranged from 0.50 to 0.85 (Figure 1). The ordinal alpha coefficient by dimension was 0.80 for the “Stressors” factor and 0.78 for the “Baby’s Rewards” factor. In terms of convergent validity, scores on the PSS scale’s “Baby’s Rewards” factor correlate positively with the STAI E-7 (r = 0.152, p < 0.05). However, no statistical association was found with the “Stressors” factor (r = 0.117; p = 0.055). In the criterion variable of perceived anxiety, the PSS correlates positively for both the “Baby’s Rewards” factor (r = 0.218; p < 0.01) and the “Stressors” factor (r = 0.197; p < 0.01).”